

# Investigation of euthanasia techniques in four species of cockroaches

Samuel M. Tucker[1,2], Julie A. Balko[1,3], Dustin C. Smith[3], Larry J. Minter[1,3] and Emma L. Houck[1,4]

[1] College of Veterinary Medicine, North Carolina State University, Raleigh, NC, United States of America
[2] Veterinary Corps, U.S. Army, APO, AE, United States of America
[3] North Carolina Zoo, Asheboro, NC, United States of America
[4] College of Veterinary Medicine, Cornell University, Ithaca, NY, United States of America

## ABSTRACT

While cockroaches are commonly exhibited in zoos and museums, studied in research laboratories, and even kept as pets, scientifically based guidelines for their euthanasia are lacking. This study assessed euthanasia techniques in four species of cockroaches (Dubia (*Blaptica dubia*), red runner (*Shelfordella lateralis*), Madagascar hissing (*Gromphadorhina portentosa*), and giant cave (*Blaberus giganteus*)). In an initial pilot study, two hundred fifty adult Dubia cockroaches were exposed in groups of ten to a cotton ball soaked with 2 mL of isoflurane in a 1 L air-tight chamber. Thirty minutes beyond loss of any individual movement, groups were exposed to one of the following secondary treatments: freezing at $-18\,°C$ or $-80\,°C$ from 0.25 to 24 hours; immersion in 10% neutral buffered formalin, 70% isopropyl alcohol, or reverse osmosis water for 0.25 or 0.5 hours; or intracoelomic injection of potassium chloride (456 mEq/kg) or pentobarbital-based euthanasia solution (3.9 g/kg). A control group remained in the air-tight isoflurane chamber for 24 hours. Following all treatments, cockroaches were monitored for an additional 24 hours for spontaneous movement. Irreversible loss of movement was considered synonymous with irreversible loss of consciousness (death). Across all species, isoflurane anesthesia followed by either 70% isopropyl alcohol immersion for 0.25 or 0.5 hours or isoflurane exposure for 24 hours resulted in euthanasia in 100% of cockroaches. This study is the first evaluation of American Veterinary Medical Association-recommended euthanasia protocols in cockroaches.

## INTRODUCTION

Cockroaches are commonly encountered in zoological institutions and natural history museums as collection animals, food sources, and research subjects, along with being found in households as pets or pests. With the increasing popularity of cockroaches and other terrestrial invertebrates in managed care, along with growing literature concerning pain perception in these species, there has been heightened interest in their welfare, including their veterinary care (*Cooper, 2012*; *Drinkwater, Robinson & Hart, 2019*; *Elwood, 2011*; *Murray, 2012*). Relatedly, euthanasia is a common procedure for this taxon (*e.g.*, research endpoint, quality of life decision, depopulation) and ensuring a humane death, free

Corresponding author
Emma L. Houck, ehouck@cornell.edu

of aversive behavior, is of utmost importance and is an integral component of both animal welfare and veterinary medicine. Currently, there is limited prospective research investigating euthanasia methods in invertebrates, including cockroach species.

The 2020 American Veterinary Medical Association (AVMA) Guidelines for the Euthanasia of Animals recommends several methods for invertebrate euthanasia including both one-step and two-step techniques (*Leary et al., 2020*). One-step techniques include anesthetic overdose using either injectable or inhaled drugs, while two-step techniques include inhalant anesthesia followed by a secondary chemical or physical method, such as immersion in alcohol or formalin, boiling, freezing, or pithing. Of importance, as the latter techniques (*e.g.*, immersion in solution) could impart prolonged aversive behavior and potentially distress or suffering, they are only recommended for use as part of a two-step technique in animals that have already been rendered unconscious as the first step (*Gilbertson & Wyatt, 2016*; *Pizzi, 2006*). Despite the existence of terrestrial invertebrate euthanasia recommendations by the AVMA, minimal species-specific objective data truly supports these recommendations.

A single prospective investigation of cockroach euthanasia exists (*Bennie et al., 2012*). In that study, giant cockroach (*Blaberus giganteus*) nymphs were anesthetized with carbon dioxide ($CO_2$) and then injected with potassium chloride (KCl) at concentrations of 1% (40 mEq/kg), 5% (201 mEq/kg), or 10% (402 mEq/kg) volume per weight. The highest dose was 100% effective for euthanasia and, while adults were not included in the study, the positive result supports further investigation of this method in other cockroach species and life stages.

Inhalant anesthetics are both anecdotally used and recommended in both one-step and two-step invertebrate euthanasia techniques, despite minimal to no prospective investigation for this purpose. Studies have described inhalant anesthetic safety and efficacy in certain invertebrate taxa, including a study in the Madagascar hissing cockroach, but overdose exposure has not been investigated (*D'Ovidio, Monticelli & Adami, 2021*; *MacMillian et al., 2017*; *McCallion et al., 2021*; *Zachariah et al., 2009*).

Freezing has also been anecdotally used as a method of achieving unconsciousness or as a one-step euthanasia method. As such, research has evaluated the physiologic response of invertebrates to colder temperatures (*Bradt, Hoback & Kard, 2018*; *Hamilton, 1985*; *Lubawy et al., 2019*; *Lubawy & Słocińska, 2020*; *Sinclair, 1997*; *Toxopeus & Sinclair, 2018*; *Worland, Wharton & Byars, 2004*). While there is support for its use in immobilizing invertebrates for non-invasive procedures, including photography and radiograph acquisition, the present study included freezing as part of a two-step method of euthanasia following induction of anesthesia (*Cooper, 2011*; *Cooper, 2012*).

Immersion in various solutions has also been investigated and is a current method recommended for euthanasia of invertebrate species (*Dombrowski & De Voe, 2007*; *Gilbertson & Wyatt, 2016*; *Murray, 2012*). Recommended solutions include 70% ethanol or 10% formalin (*Dombrowski & De Voe, 2007*; *Murray, 2012*), with the former used in situations of desired postmortem analysis as tissues are preserved (*Dombrowski & De Voe, 2007*). Currently, there is limited literature on the use of these techniques in cockroach species.

The objective of this study was to evaluate euthanasia techniques in four species of cockroaches (Dubia (*Blaptica dubia*), red runner (*Shelfordella lateralis*), Madagascar hissing (*Gromphadorhina portentosa*), and giant cave (*Blaberus giganteus*)), with inclusion of techniques recommended for invertebrates by the AVMA Guidelines for the Euthanasia of Animals. It was hypothesized that effective techniques would include: prolonged exposure (≥24 h) to isoflurane inhalant, anesthesia with isoflurane followed by injection with potassium chloride or pentobarbital-based euthanasia solution, and anesthesia with isoflurane followed by immersion in isopropyl alcohol, neutral buffered formalin, or reverse osmosis water for ≤30 min.

## MATERIALS & METHODS

### Animals

To minimize extraneous testing, all techniques were first assessed in only a single cockroach species (Dubia), which included 260 individuals. Effective techniques were then tested on three additional cockroach species, which included 70 red runner, 70 Madagascar hissing, and 60 giant cave cockroaches. These species were selected based upon their commonality as pets or presence in zoological institutions as education or display animals (*G. portentosa, B. giganteus*) or as feeder colonies (*B. dubia, S. lateralis*). Age was not controlled but animals in all groups were identified as sexually mature (*i.e.,* adult) based on external morphology. Cockroaches were group housed by species in plastic containers with newspaper substrate and a hide in a climate-controlled environment (humidity 65–75%, temperature 75–80 °F). Animals were fed a variety of fruits, vegetables, Natural Balance Limited Ingredient Salmon & Sweet Potato formula dog food (Natural Balance Pet Foods, Inc, Upland, CA, USA), and Soil Moist (JRM Chemical, Inc,. Cleveland, OH, USA) daily. Prior to testing, each cockroach received a brief visual examination to ensure normal response to environment and mobility. All testing was performed in a climate-controlled room (75–80 °F) unless otherwise noted. This study was approved by the North Carolina Zoo Research Committee and performed in agreement with recommended invertebrate medicine guidelines to ensure appropriate animal welfare (*Bennie et al., 2012*; *Leary et al., 2020*).

### Experimental protocol
#### Pilot study

As isoflurane anesthesia in Dubia cockroaches had not been characterized at the time of this study, a pilot study was undertaken to guide its use as the first step in subsequent two-step techniques; the specific objective was identification of an anesthetic protocol that provided the longest duration of anesthesia with no mortality. Ten Dubia cockroaches were placed in a 1 L air-tight plastic container (LockNLock; Seocho, Seoul, South Korea) modified to connect to the fresh gas outlet of an isoflurane vaporizer (Tec 3) and passive waste gas scavenger system (F/air canister; A.M. Bickford, Inc., Wales Center, NY, USA). Cockroaches were exposed to 5% isoflurane (Isoflurane USP; Piramal Critical Care, Telangana, India) delivered in 100% oxygen ($O_2$) at 1 L/min and monitored for loss of spontaneous movement of all individuals. After 15 s of absent spontaneous movement, the container was gently rocked at an approximately 30° angle along its short axis in both a left

and right direction and cockroaches were monitored for spontaneous movement of any individual. If movement was present, cockroaches were left unstimulated and this process was repeated 1–2 min following cessation of movement. If movement was absent, the time was recorded and considered synonymous with group anesthetic induction. Cockroaches were left undisturbed in the container for an additional 5 min to allow for peak anesthetic effects, after which, isoflurane and oxygen delivery were discontinued. Cockroaches were then removed from the container and monitored for return of spontaneous movement of the first individual cockroach which was designated as group recovery. To allow flexibility for use in clinical settings and assess for response to a higher concentration of inhalant anesthetic, an additional method of isoflurane delivery was also assessed. Twenty naive Dubia cockroaches (n = 10/group) were placed in a 1 L air-tight plastic container with a cotton ball soaked with two mL of liquid isoflurane as previously described in invertebrate anesthesia guidelines (*Braun, Heatley & Chitty, 2006*). As one mL of liquid isoflurane can produce 195 mL of vapor (*Biro, 2014*), this results in a maximum isoflurane concentration of 32% (saturated vapor pressure) within the container. Monitoring and assessment proceeded as previously described. Following group anesthetic induction, cockroaches were left unstimulated in the container for an additional either 5 or 30 min. Cockroaches were then removed from the container and monitored for group recovery, as defined above. Of the three pilot test groups, exposure to liquid isoflurane *via* cotton ball for 30 min beyond anesthetic induction provided the longest duration of anesthesia with complete recovery of all individuals. Thus, this was selected as the first step technique for use in the principal study.

Exposure to carbon dioxide was also assessed as an alternative first step technique, but proved unsuccessful. Cockroaches displayed aversive behaviors during the induction period (rapid and erratic antennae movement, appendage tremoring, body curling or tucking, exaggerated, repeated coelomic expansions) and recovered within 5 min of induction, regardless of exposure time.

### Test groups
Four hundred sixty naive cockroaches of four species were enrolled in a controlled study assessing isoflurane exposure for 24 h (one step) and isoflurane anesthesia (first step) followed by one of the following second steps: freezing at $-18\,°C$ or $-80\,°C$, both for 0.25, 0.5, 1, 2, 4, 8, 16, or 24 h; immersion in 10% buffered formalin, 70% isopropyl alcohol, or reverse osmosis water for 0.25 or 0.5 h; or intracoelomic injection of potassium chloride or pentobarbital-based euthanasia solution.

## Two step techniques
### First step (anesthesia)
Twenty-four cohorts of naive Dubia cockroaches (n = 10/group) were group exposed to a cotton ball soaked with two mL of liquid isoflurane in a 1 L air-tight plastic container. The container was modified with two air-tight ports each created using a 20-gauge x 2.9 cm catheter (BD Insyte, Covetrus, Columbus, OH, USA) and attached injection cap (ICU Medical, Inc., San Clemente, CA, USA). During exposure, cockroaches were monitored for movement as described in the pilot study. An additional cotton ball soaked with

two mL of liquid isoflurane was added to the container for any groups that exhibited spontaneous movement at 1 h beyond initial exposure. At induction, cockroaches were left unstimulated for an additional 30 min to allow peak anesthetic effects. A multi-gas analyzer (Mindray, Passport 12, Mindray, Mahwah, NJ, USA) sampling and waste line was briefly and simultaneously attached *via* a 22-gauge ×2.5 cm needle (Monoject™, Cardinal Health, Dublin, OH 43017, USA) to each container port at 5-minutes of exposure, anesthetic induction, and end-exposure for collection of $O_2$, $CO_2$, and isoflurane concentrations. At end-exposure, cockroaches were removed from the container and individually weighed, sexed, and assessed for spontaneous movement or response to manual manipulation. If absent for all individuals, the group randomly received a second step technique as described below.

### Second step (freezing, immersion, or injection)

Following exposure to isoflurane for 30 min beyond anesthetic induction as described above, each group was exposed to one of three techniques. For freezing ($n = 16$ groups), cockroaches were placed in dorsal recumbency in individual compartments ($4.2 \times 3.0 \times 4.3$ cm or $4.4 \times 4.9 \times 5.0$ cm) within clear, plastic, vented containers (Compartment Box, The Container Store, Coppell, TX, USA) in one of two commercial-grade freezers ($-18\ °C$ or $-80\ °C$) for a total of 0.25, 0.5, 1, 2, 4, 8, 16, or 24 h. For immersion ($n = 6$ groups), cockroaches were group-immersed in 100 mL of 10% neutral buffered formalin (Thermo Fisher Scientific Inc., Waltham, MA, USA), 70% isopropyl alcohol (First Priority, Inc., Elgin, IL, USA), or reverse-osmosis water in a 500 mL sealed, cylindrical plastic container and for 0.25 or 0.5 h. Volume of solution was chosen to ensure coverage of spiracles and containers were gently agitated following addition of cockroaches. For injection ($n = 2$ groups), cockroaches were individually, manually restrained in sternal recumbency and injected intracoelomically along midline with potassium chloride (456 mEq/kg) (340 mg/mL; ACS Reagent 99.0–100.5%, Sigma-Aldrich, St. Louis, MO, USA) or pentobarbital-based euthanasia solution (3.9 g/kg) (390 mg/mL pentobarbital sodium, 50 mg/mL phenytoin sodium; Euthanasia Solution, Vedco Inc., Saint Joseph, MO, USA) using a 25-gauge × 1.56 cm needle and one mL tuberculin syringe (Monoject™; Cardinal Health, Dublin, OH, USA). Injection targeted the thoracic ganglia as described by Bennie et al. and doses were extrapolated from previous invertebrate literature and rounded to the nearest 0.01 mL (*Bennie et al., 2012*; *Leary et al., 2020*).

Following treatment in all groups, cockroaches were individually assessed for spontaneous movement or response to manual manipulation. If absent, cockroaches were placed in dorsal recumbency in individual compartments ($4.2 \times 3.0 \times 4.3$ cm or $4.4 \times 4.9 \times 5.0$ cm) within clear, plastic, vented containers. Cockroaches were left unstimulated for 24 h and then individually reassessed for spontaneous movement, response to manual manipulation, and/or ability to right themselves. If all were absent, euthanasia was considered successful. Any cockroach that demonstrated spontaneous movement, response to manual manipulation, or any other indicator of resumption of consciousness during the second step or recovery period was anesthetized *via* isoflurane exposure as described above and macerated *via* a blender (Ninja Professional Plus™,

SharkNinja Operating LLC, Needham, MA, USA) to ensure complete disruption of the invertebrate ganglia (*Dailey & Graves, 1976*; *Gullan & Cranston, 1994*).

## One step technique
### *Isoflurane Exposure for 24 hours*

Ten naive Dubia cockroaches were group exposed to a cotton ball soaked with 2 mL of liquid isoflurane in a 1 L air-tight plastic container modified with two air-tight ports. Monitoring and assessment were performed as described above. Following group induction, cockroaches were left unstimulated for 24-hours. Collection of intra-container $O_2$, $CO_2$, and isoflurane concentrations was performed as previously described at induction and 1, 17, and 24 h beyond induction. At end-exposure, cockroaches were individually weighed, sexed, and assessed for spontaneous movement or response to manual manipulation. If the latter were absent, cockroaches were placed in dorsal recumbency in individual containers and monitored for recovery, as previously described.

### *Control*

In each two-step technique, isoflurane exposure was continued for 30 min beyond anesthetic induction, however, this time point and subsequently total isoflurane exposure time was variable between groups. To assess this potential confounding factor, ten cockroaches of each species were group exposed to isoflurane for the longest exposure duration documented for each species in the previous trials. Exposure, monitoring and assessment, including collection of intra-container gas concentrations at end-exposure and monitoring for movement and response to manipulation following exposure, were performed as previously described. Time to group recovery was recorded.

### *Method expansion*

Techniques that were ≥90% (injectable) or 100% (all other techniques) successful in Dubia cockroaches were repeated in red runner, Madagascar hissing, and giant cave cockroaches. A 90% cutoff was chosen for injectable methods as these techniques have subjectively greater potential for inter-individual variability in response (*e.g.*, technique failure, leakage from injection site). To minimize animal usage, only the minimum effective exposure duration for freezing or immersion was tested. Sample size limitations precluded isoflurane exposure for 24 h in giant cave cockroaches. Methodology for each trial was performed as described above for Dubia cockroaches.

## Statistical analysis

Data was tested for normality using a Shapiro Wilk normality test and then analyzed *via* one-way analysis of variance (ANOVA) or Kruskal–Wallis test with either Tukey's or Dunn's post-hoc test used for multiple comparisons, respectively. Data analyses were performed with a commercially available statistical software (Prism 7.0; GraphPad Software Inc, La Jolla, CA, USA) and significance was set at $P < 0.05$.

## RESULTS

Overall median (range) body weight amongst test groups was 2.15 (1.1–3.6), 0.4 (0.1–1.5), 5.35 (1.8–11.1), and 6.1 (3.7–7.4) grams for Dubia ($p = 0.1565$), red runner ($p = 0.8482$),

Madagascar hissing ($p = 0.5338$), and giant cave ($p = 0.0643$) cockroaches, respectively, and was not significantly different within each species. Overall, the male:female ratio for all principal test groups was 0.73:1 0.4:1, 0.84:1, and 0.41:1 for Dubia, red runner, Madagascar hissing, and giant cave cockroaches, respectively.

Median (range) time to anesthetic induction amongst all groups in the principal study was 25.5 (12.8–35.7), 15.8 (8.0–22.5), 43.3 (26.8–50.0), and 66.7 (45.3–80.8) minutes for Dubia, red runner, Madagascar hissing, and giant cave cockroaches, respectively; this was significantly different between all groups ($p < 0.0001$). An additional isoflurane-soaked cotton ball was needed for two giant cave cockroach exposure groups (70% alcohol immersion for 0.25 h and isoflurane exposure for 24 h). Six giant cave cockroaches responded to manipulation prior to administration of their target second step technique, thus, they were removed from the study and euthanized as previously described. Median (range) isoflurane concentration at induction was 19.7 (12.8–30.0), 13.3 (11.9–30.0), 18.9 (16.0–22.8), and 18.7 (15.3–30.0) % for the Dubia, red runner, Madagascar hissing, and giant cave cockroaches, respectively; this was not significantly different between groups ($p = 0.0891$). Isoflurane concentration at end-exposure (ie, 30-minutes post-induction) was 30% (the limits of monitor detection) for 23/34, 6/7, 2/7, and 3/6 groups amongst Dubia, red runner, Madagascar hissing, and giant cave cockroaches, respectively. Median (range) $CO_2$ concentration at end-exposure was 4 (0–6), 2 (0–2), 7 (2–8), and 8 (6–9) mmHg for Dubia, red runner, Madagascar hissing, and giant cave cockroaches, respectively. For the 24-hour isoflurane exposure, isoflurane concentrations remained at 30% (the limits of monitor detection) until end-exposure for Dubia and red runner cockroaches, but, in Madagascar hissing cockroaches, concentrations fell to 16.14% in 17 h and 11.22% in 24 h, a 46.2% and 62.6% decrease, respectively.

Median (range) potassium chloride injection volume for Dubia, red runner, Madagascar hissing, and giant cave cockroaches was 0.17 (0.12–0.26), 0.03 (0.02–0.06), 0.63 (0.19–1.09), and 0.64 (0.56–0.74) mL, respectively. Median (range) pentobarbital-based euthanasia solution injection volume for Dubia, red runner, Madagascar hissing, and giant cave cockroaches was 0.01 (0.01–0.03), 0.01 (0.01), 0.08 (0.03–0.10), and 0.07 (0.05–0.07) mL, respectively.

Euthanasia success rates are reported in Table 1. In the control groups, all exposed cockroaches amongst all four species recovered. Aside from euthanasia, no other clinically apparent adverse effects were noted in any subject in any group. Of note, several females across all species expelled oothecas during the study.

## DISCUSSION

Across all species, isoflurane anesthesia followed by either 70% alcohol immersion for 0.25 or 0.5 h (Dubia only) or isoflurane exposure for 24 h resulted in euthanasia in 100% of cockroaches. Isoflurane anesthesia followed by freezing or injection with either pentobarbital-based euthanasia solution or potassium chloride were 100% effective for some groups, but this varied by species and/or exposure duration. Cockroach-specific recommended euthanasia techniques based on the results of this study are presented in Table 2.

**Table 1  Results of investigated euthanasia techniques in four species of cockroaches.** Euthanasia techniques with duration periods/dosages explored across all four species of cockroaches with mortality (%) calculated from number per group. * = 0–69%, Italicized = 70–99%, Bold = 100%.

| Treatment | | Mortality (%) by species | | | |
| --- | --- | --- | --- | --- | --- |
| | | *B. dubia* | *S. lateralis* | *G. portentosa* | *B. giganteus* |
| **Freezing** | **Duration (h)** | | | | |
| −18C | 0.25 | 10* | | | |
| | 0.5 | 10* | | | |
| | 1 | **100** | **100** | 20* | *90* |
| | 2 | **100** | | | |
| | 4 | **100** | | | |
| | 8 | **100** | | | |
| | 16 | **100** | | | |
| | 24 | **100** | | | |
| −80C | 0.25 | *70* | | | |
| | 0.5 | **100** | **100** | *90* | **100** |
| | 1 | **100** | | | |
| | 2 | **100** | | | |
| | 4 | **100** | | | |
| | 8 | **100** | | | |
| | 16 | **100** | | | |
| | 24 | **100** | | | |
| Immersion | Duration (h) | | | | |
| Water | 0.25 | 20* | | | |
| | 0.5 | 10* | | | |
| Formalin | 0.25 | 0* | | | |
| | 0.5 | 40* | | | |
| Alcohol | 0.25 | **100** | **100** | **100** | **100** |
| | 0.5 | **100** | | | |
| Injection | Dosage | | | | |
| Potassium chloride (mEq/kg) | 456 | *90* | **100** | **100** | **100** |
| Pentobarbital (g/kg) | 3.9 | **90** | **100** | *90* | **100** |
| Inhalant overdose | Duration (h) | | | | |
| Isoflurane | 24 | **100** | **100** | **100** | |

Euthanasia *via* isoflurane anesthesia followed by short-duration immersion in 70% isopropyl alcohol is consistent with previous recommendations for terrestrial invertebrate immersion euthanasia (*Dombrowski & De Voe, 2007*; *Gilbertson & Wyatt, 2016*; *Pizzi, 2006*). This technique is technically simple and requires minimal supplies, which are readily and widely accessible. In contrast to the other effective technique in this study (prolonged isoflurane exposure), 70% isopropyl alcohol immersion of anesthetized cockroaches necessitates a significantly shorter time commitment and, thus, may be more practical in a clinical setting. It is unknown if exposure times <0.25 h would result in euthanasia.

It was hypothesized that immersion of anesthetized cockroaches in test solutions would inundate the respiratory system, impede oxygen uptake, and result in hypoxia and death. This was documented in the head louse (*Pediculosis humanus capitis*) whereby

**Table 2  Recommended and not recommended euthanasia techniques.** The recommended and not recommended euthanasia techniques for cockroach species based on the results of this study.

| Treatments | Recommended duration/dosage by species | | | |
| --- | --- | --- | --- | --- |
| | *B. dubia* | *S. lateralis* | *G. portentosa* | *B. giganteus* |
| Freezing at −18C (h) | >1 | >1 | NR[a] | NR[a] |
| Freezing at −80C (h) | >0.5 | >0.5 | NR[b] | >0.5 |
| Alcohol immersion (h) | >0.25 | >0.25 | >0.25 | >0.25 |
| Potassium chloride (mEq/kg) | NR | 456 | 456 | 456 |
| Pentobarbital (g/kg) | NR | 3.9 | NR | 3.9 |
| Inhalant overdose (h) | >24 | >24 | >24 | NE |

Notes.
NR, Not recommended; NE, Not evaluated.
[a] Only evaluated treatment duration of 1 h.
[b] Only evaluated treatment duration of 0.5 h.

exposure to benzyl alcohol lotion prevented closing of the spiracle and the above-described pathogenesis (*Meinking et al., 2010*). However, if this hypothesis were true, all three tested solutions should have demonstrated equal efficacy, which was not the case in the current study. Alternatively, it is plausible that the chemical makeup of 70% isopropyl alcohol was responsible for irreversible unconsciousness, but the ultimate mechanism is unknown. To the authors' knowledge, no research has investigated the efficacy of immersion in reverse osmosis water and this technique was not effective in Dubia cockroaches. Similarly, while a single study demonstrated effectiveness of immersion in 10% neutral buffered formalin for the euthanasia of ethanol-anesthetized land snails (*Succinea putris*) (*Braun, Heatly & Chitty, 2006*), this technique was not effective in the current study.

Prolonged (>24 h) exposure to isoflurane was also effective for euthanasia of Dubia, red runner, and Madagascar hissing cockroaches in the current study; recall that giant cave cockroaches were not tested due to sample size limitations. This technique was subjectively easy to implement, required minimal supplies, and necessitated only a single step. Interestingly, invertebrate guidelines widely recommend inhalant anesthesia (isoflurane, sevoflurane) for immobilization of terrestrial arthropods, however, prior to the current study, no investigation into the effects of prolonged exposure had been performed. Anecdotally, mortality is low following inhalant exposure (*Cooper, 2001*; *Cooper, 2011*; *Dombrowski & De Voe, 2007*). Target isoflurane exposure concentration (30%) was equivocally reached in all groups but there was a decrease in concentrations observed in the Madagascar hissing cockroaches in the prolonged (>24 h) isoflurane exposure. This may have been the result of increased inhalant uptake in a larger cockroach species or due to a leak in the container. A repeat trial with the same species or one of similar size could help elucidate this etiology.

The effectiveness of prolonged exposure of isoflurane-anesthetized cockroaches to freezing temperatures in the current study was both species and exposure duration dependent. Exposure to −18 °C or −80 °C for ≥1 h or ≥30 min, respectively, was 100% effective for euthanasia of Dubia cockroaches in the current study. Of the remaining three species, red runner cockroaches were the only other species with 100% effectiveness at both

temperatures. However, as only a single exposure duration was tested in these species, results should be interpreted as preliminary and caution practiced accordingly. There is extensive literature regarding anecdotal usage of freezing for immobilization and euthanasia of cockroaches and other invertebrates (*Bradt, Hoback & Kard, 2018*; *Braun, Heatly & Chitty, 2006*; *Cooper, 2012*; *Lubawy et al., 2019*; *Lubawy & Słocińska, 2020*). Because the experience of cockroaches is unknown and difficult or impossible to interpret, especially during the freezing process, this study intentionally used a two-step method with anesthesia prior to freezing to reduce the chance that cockroaches had sensation during the freezing process. While this study did not directly evaluate freezing as a one-step method, it is likely that the effectiveness of freezing in anesthetized cockroaches is similar to or greater than unanesthetized cockroaches in a one-step euthanasia process, but further studies of success rates are necessary. Given that freezing is a common one-step method of euthanizing cockroaches in a variety of settings and that a two-step method may risk exposure of personnel to anesthetic gases, further studies may provide useful comparison between anesthetized and non-anesthetized cockroaches undergoing freezing. There is much controversy regarding the experiences of animals undergoing fatal hypothermia, including reptiles and invertebrates. This study did not evaluate the experiences of cockroaches during freezing but simply reported the success rates of each method, which was shown to vary by species.

Given the variety of temperature extremes that cockroaches naturally inhabit, there has been further investigation into possible freeze tolerance in invertebrate species. It has been suggested that freeze-tolerant insects can control the quantity and quality of ice formation within their exoskeletons by disrupting the cellular dehydration and mechanical destruction that freezing temperatures can induce (*Toxopeus & Sinclair, 2018*). One study demonstrated the physiological durability of Madagascar cockroaches (*Gromphadorinha coquereliana*) to tolerate colder temperatures ($-4.76\,°C$), which was consistent with findings observed in this study (*Lubawy et al., 2019*). Other documented freeze tolerant cockroach species include the New Zealand alpine cockroach (*Celatoblatta quinquemaculata*) and the brown-hooded cockroach (*Cryptocercus punctulatus*) (*Sinclair, 1997*; *Hamilton, 1985*); *Worland, Wharton & Byars, 2004*). In contrast, the American cockroach (*Periplaneta americana*) was unable to survive temperature extremes (*Bradt, Hoback & Kard, 2018*). Based upon the previously described evidence of freeze tolerance across different species and the different efficacy in treatment groups observed in this study, freezing may be an effective method of euthanasia in some species of cockroach but not in others. Further investigation into the effect of different durations of freezing temperatures in cockroach species is recommended, especially given that this method is frequently utilized as both a one-step and two-step method of euthanasia.

Efficacy of pentobarbital-based euthanasia solution and potassium chloride in the current study were 90% or greater for all tested species. As each test group only had 10 individuals, this translates to technique failure in $\leq 1$ cockroach amongst each group. While the etiology of failure is unknown, it is possible that technical error (drug leakage from injection site, injection failure) may have played a role. These findings are consistent

with previous evidence-based studies investigating injectable euthanasia techniques in invertebrate species (*Battison et al., 2000*; *Bennie et al., 2012*; *Heniff et al., 2023*).

While an injectable method of euthanasia may be appropriate for individual animals, there are potential downsides to this technique. First, injection may not be efficient or clinically practical for a large cohort of cockroaches which often present as part of research or feeder colonies. Additionally, injectable drugs may leave tissue residues that limit subsequent use of the tissue as a foodstuff. Finally, if postmortem analysis is desired, an injectable method may negatively affect histopathology results. Histopathologic analysis was outside the scope of the current study, but in a study of American cockroaches (*Periplaneta americana*), organic solvents including acetone, hexane, and chloroform affected gross pathology results (*Patton & Sakaria, 1958*).

Body weight can positively or negatively influence induction time of topical anesthetics in fish, amphibians, and aquatic invertebrates, but this has not been documented in terrestrial invertebrates (*Li et al., 2020*; *Popovic et al., 2012*; *Zec et al., 2014*). In the current study, median time to anesthetic induction was over four times faster in red runner cockroaches (species of a smaller body weight) compared to giant cave cockroaches (species of larger body weight). In contrast, American green tree frogs (*Dryophytes cinereus*) of a larger body weight had a faster induction time than those of a smaller body when exposed to topical isoflurane (*Zec et al., 2014*). The significance of this finding in the current study is unknown, but explanations may include differences in anatomy or physiology, dosing requirements, or true effects of body weight.

Aside from the expulsion of ootheca in several females across different treatment cohorts, no significant aversive behaviors were observed in this study. Possible explanations for ootheca protrusion include behavioral stress, muscle relaxation, or a combination. Given the lack of other signs characteristic of distress (muscle tremors, hyperexcitability, spiracle contraction), this is likely a result of muscle relaxation.

Several limitations were encountered in the current study. Namely, there was a lack of sufficient number of clinically applicable indicators of invertebrate unconsciousness and death, small sample sizes, and the use of only four cockroach species despite the diversity of this taxon. Measurement of heart rates is commonly used to confirm successful euthanasia, however, while cardiac monitoring has been described, it has been determined to be unfeasible in the majority of invertebrate species, including the species in this study (*Battison et al., 2000*; *Coelho & Amaya, 2000*; *Cooper, 2012*). Additionally, there are no standardized assessments for cockroach consciousness or pain.

## CONCLUSIONS

In conclusion, this study provides a comprehensive and evidence-based assessment of euthanasia techniques in Dubia, red runner, Madagascar hissing, and giant cave cockroaches. As such, prolonged (24 h) exposure to isoflurane or isoflurane anesthesia followed by short duration ($\geq 0.25$ h) immersion in 70% isopropyl alcohol may be considered for euthanasia of cockroach species. This study provides the first evidence-based assessment of euthanasia techniques across several different species of cockroach.

## ACKNOWLEDGEMENTS

The authors would like to thank the streamside keepers and ambassador animal staff of the North Carolina Zoo for their assistance in the care of these cockroaches.

### Funding

The authors received no funding for this work.

### Competing Interests

The authors declare there are no competing interests.

### Author Contributions

- Samuel M. Tucker conceived and designed the experiments, performed the experiments, analyzed the data, prepared figures and/or tables, authored or reviewed drafts of the article, and approved the final draft.
- Julie A. Balko conceived and designed the experiments, performed the experiments, analyzed the data, prepared figures and/or tables, authored or reviewed drafts of the article, and approved the final draft.
- Dustin C. Smith conceived and designed the experiments, authored or reviewed drafts of the article, and approved the final draft.
- Larry J. Minter conceived and designed the experiments, performed the experiments, authored or reviewed drafts of the article, and approved the final draft.
- Emma L. Houck conceived and designed the experiments, analyzed the data, prepared figures and/or tables, authored or reviewed drafts of the article, and approved the final draft.

### Data Availability

The raw data is available in the Supplemental File.

### Supplemental Information

Supplemental information for this article can be found online at http://dx.doi.org/10.7717/peerj.16199#supplemental-information.

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
