# Peer review of "Investigation of euthanasia techniques in four species of cockroaches"

_PeerJ, doi:10.7717/peerj.16199_

## Round 0.1 · original submission · Major Revisions

Thank you for submitting this manuscript to PeerJ. Your manuscript has been reviewed by two experts in the field, and overall they found your manuscript to be well-written, convincing and important.

I generally agree with Reviewer 2, in that you really should be evaluating freezing alone as a method of humane killing for cockroaches, because that is standard protocol for many invertebrates. If you cannot test freezing alone, you need to provide a strong rationale for why not, with a statement saying that it is still an acceptable mode of humanely killing these animals.

Please see and respond to the reviewer's comments in your next submission and provide a detailed rebuttal letter indicating where you have made the corresponding changes in the manuscript.

Thank you and I look forward to reading the next submission.

·

Basic reporting

Introduction and background clearly define the context the authors are working in and provide an appropriate presentation of the study. References are complete and relevant.
Structure conforms to PeerJ standards.
The Research Proposal Form for North Carolina Zoo has been duly filled in and has been approved by the NCZ IACUC (Institutional Animal Care and Use Committee). All relevant information are included.

.

Experimental design

This is an original primary research that falls within Scope of the journal.
The goal of the research is well defined, relevant and justified; it is clearly stated that this research will fill some knowledge gaps about euthanasia in four cockroaches species.
The experiment protocol is rigorously described with all needed details.
Ethical standards are carefully considered in all steps of the protocol.

Validity of the findings

The novelty of the study is stated and the practical impact of its results are properly supported.
Statistical analysis supports the structure of the study and the validity of the results obtained.
Tables clearly summarizes the results obtained in the study.

Additional comments

The article is well written, clear, easily readable and understandable.
Line 339: "and" should be probably deleted

·

Basic reporting

Good except for the recommendation not to use freezing given limitations of exposure and study design (See comments below).

Experimental design

Since freezing adult cockroaches in a commissary-kitchen freezer at 0 degrees F (-18C) is the most common euthanasia method used in zoos and museums, the authors should evaluate this method as a one-step method until 100% of all species are dead. Ideally, the cockroaches should be observed during the freezing process for aversive behaviors. Recognizing real time observations may not be feasible, the authors should at least check for unconsciousness/ death/ no movement at one-hour intervals as done in Table 1 as well as lack of recovery at room temp . Freezing should be also similarly be evaluated for all species as a secondary method at 30 minute to 1-hour intervals before “not recommending freezing” as an acceptable primary or secondary method of euthanasia.

Validity of the findings

LOTS of excellent tarcking of microenvironmental parameters and time to detah.

The authors should have evaluated efficacy of freezing of all 4 species over 30 minute to hourly intervals before not recommending freezing as an acceptable secondary method. The authors should also evaluate freezing as a primary method of euthanasia.

Additional comments

In the attached PDF, I've made some suggestions in blue font and deletions in red font especially when implying invertebrates experience pain or distress. Best to use "aversive behaviors". For the abstract, I also deleted reference to cockroaches in houses as readers may think the authors are implying pest cockroaches should experience a good death "euthanasia" as described in the publication.

---

## Round 0.2 · Major Revisions

Thank you for your resubmission to PeerJ.

This revision was reviewed by one reviewer who has highlighted again that there is a flaw in the experimental design or introduction of this paper. Please address the points raised about freezing as a method of humane killing.

It would be best if you can experimentally show the effects of freezing alone. If you cannot do that, then you need to make it abundantly clear in the intro and the discussion that you did not test freezing alone and therefore there is no evidence to suggest that freezing alone is inhumane.

The implication that freezing alone is inhumane will, as the reviewer stated, have an incorrect influence on institutions like animal ethics. We need to be really careful of the downstream impacts of animal welfare experiments and how that can affect animal ethics procedures and potentially harm practitioners is critical and needs to be carefully and thoughtfully addressed in this manuscript before we can approve it for publication here at PeerJ.

·

Basic reporting

clearly written. relevant references.

Experimental design

Research question flawed given no evidence that freezing as a primary method of euathansia is distressful.

Validity of the findings

Line 61-64 implying that exposure to freezing temperatures could impart prolonged aversive behavior and potentailly distress and suffering is misleading and not supported by the references listed. Freezing as a primary method of euthanasia should be evaluated before suggesting use of isoflurane risking personnel exposure to waste anesthetic gas. Publishing this manuscript implying that freezing is inhumane without ruling out freezing as an acceptable primary method of euthanasia will incorrectly influence institutions to unecessarily use isoflurane.

---

## Author Rebuttal · Round 0.2

Dear Editor,                                                    13 June 2023

We thank the reviewers for their generous comments on our manuscript. We have edited the manuscript accordingly to address their concerns. Outlined below, we have addressed each concern highlighted by the reviewers.

We believe that the manuscript is now suitable for publication in PeerJ and will provide an excellent resource as the field of invertebrate medicine continues to grow.

Sincerely and on behalf of all co-authors,

Samuel M. Tucker, DVM
Captain, US Army Veterinary Corps
Section OIC, Vilseck Veterinary Treatment Facility
CMR 411 BOX 4757
APO, AE

## Reviewer 1 (Franco Mutinelli)

### Basic reporting

**Introduction and background clearly define the context the authors are working in and provide an appropriate presentation of the study. References are complete and relevant. Structure conforms to PeerJ standards.**

**The Research Proposal Form for North Carolina Zoo has been duly filled in and has been approved by the NCZ IACUC (Institutional Animal Care and Use Committee). All relevant information are included.**

Thank you for the feedback on our introduction and background.

### Experimental design

**This is an original primary research that falls within Scope of the journal.**

**The goal of the research is well defined, relevant and justified; it is clearly stated that this research will fill some knowledge gaps about euthanasia in four cockroaches species.**

**The experiment protocol is rigorously described with all needed details.**

**Ethical standards are carefully considered in all steps of the protocol.**

We are appreciative that our protocol was well-received.

### Validity of the findings

**The novelty of the study is stated and the practical impact of its results are properly supported.**

**Statistical analysis supports the structure of the study and the validity of the results obtained.**

**Tables clearly summarizes the results obtained in the study.**

Thank you for the feedback on our data and on its presentation in the table formats.

### Additional comments

**The article is well written, clear, easily readable and understandable.**

**Line 339: "and" should be probably deleted**

Thank you for the comment. The typo on Line 339 has been deleted.

none
## Reviewer 2 (Jeffery Wyatt)

### Basic reporting
**Good except for the recommendation not to use freezing given limitations of exposure and study design (See comments below).**

Thank you for the comments. We have addressed this concern in the below comments.

### Experimental design
**Since freezing adult cockroaches in a commissary-kitchen freezer at 0 degrees F (-18C) is the most common euthanasia method used in zoos and museums, the authors should evaluate this method as a one-step method until 100% of all species are dead. Ideally, the cockroaches should be observed during the freezing process for aversive behaviors. Recognizing real time observations may not be feasible, the authors should at least check for unconsciousness/ death/ no movement at one-hour intervals as done in Table 1 as well as lack of recovery at room temp . Freezing should be also similarly be evaluated for all species as a secondary method at 30 minute to 1-hour intervals before "not recommending freezing" as an acceptable primary or secondary method of euthanasia.**

Thank you for the comments. The co-authors were all in agreement that inclusion of freezing as a method of euthanasia in this study was needed due to its commonality as a one-step method. Concerning testing freezing as a one-step method, we sought to perform all methods following anesthesia in order as we did not fully understand the efficacy of these methods across these four species of cockroaches. We aimed to minimize aversive behaviors and, therefore, made the ethical decision to anesthetize each group prior to administering their selected euthanasia method.

Concerning "not recommending freezing", in Line 357-359 of the reviewed manuscript, we discussed the limitation of our study in that only single exposure duration was tested in the expanded three species groups. Therefore, those findings would need to be interpreted with caution and as preliminary. In the subsequent paragraph, we described how there is variation in cockroach responses to different temperatures and that our results may be an additional reflection of this. However, in this study, we merely evaluated methods and durations that were 100% in one group and performed in another to demonstrate possible species differences. Ultimately, it was outside the scope of this single study to evaluate freezing as a singular method of euthanasia.

Given the frequency of hypothermia as a primary method of euthanasia in invertebrate species, we agree it would be an interesting additional study to evaluate different exposure temperatures across different durations. We have added a clarification to our discussion of freezing in the manuscript that further studies evaluating different durations and temperatures across different species is warranted and that our results are limited in this scope.

**Validity of the findings**
**LOTS of excellent tarcking of microenvironmental parameters and time to detah.**

**The authors should have evaluated efficacy of freezing of all 4 species over 30 minute to hourly intervals before not recommending freezing as an acceptable secondary method. The authors should also evaluate freezing as a primary method of euthanasia.**

Thank you for the comments. We have addressed the primary concern in the comment above. We have updated Table 2 to better reflect that we are not recommending the evaluated temperature and duration in these species but that this method may be effective at different temperatures and durations.

**Additional comments**
**In the attached PDF, I've made some suggestions in blue font and deletions in red font especially when implying invertebrates experience pain or distress. Best to use "aversive behaviors". For the abstract, I also deleted reference to cockroaches in houses as readers may think the authors are implying pest cockroaches should experience a good death "euthanasia" as described in the publication.**

Thank you for the recommended suggestions to our manuscript. We have documented these suggestions in both the track changes manuscript and the revised manuscript. We agree that as pain or distress has not been officially described in invertebrates, use of the term "aversive behaviors" encompasses the clinical indicators of discomfort and stress that we were attempting to document. Thank you for the comment regarding pest cockroaches. We were attempting to describe the diversity of settings that individuals can encounter these invertebrates.

---

## Round 0.3 · accepted · Accept

Thank you for making those adjustments in your revision.

I believe that you have successfully addressed the reviewer's concerns regarding the lack of comparison to freezing alone. It is now clear that your study does not comment on the efficacy of freezing alone and therefore cannot and should not be used as evidence against freezing alone as a method of euthanasia.

While this study does not address the entire question of best practice for cockroach euthanasia, it does represent a worthwhile study and deserves to be published. Therefore the caveats as stated above and within the body of your manuscript are necessary for publication.

Because you have taken the reviewer's concerns into account and clearly outlined the limitations of the study I now believe that this paper is ready for publication. Congratulations!